

# Genome sequencing and analysis of *Salmonella enterica* subsp. *enterica* serovar Stanley UPM 517: Insights on its virulence-associated elements and their potentials as vaccine candidates

Khalidah Syahirah Ashari[1], Najwa Syahirah Roslan[2],
Abdul Rahman Omar[2,3], Mohd Hair Bejo[2,3], Aini Ideris[2,4] and
Nurulfiza Mat Isa[1,2]

[1] Department of Cell and Molecular Biology, Faculty of Biotechnology and Biomolecular Sciences, Universiti Putra Malaysia, Serdang, Selangor, Malaysia
[2] Institute of Bioscience, Universiti Putra Malaysia, Serdang, Selangor, Malaysia
[3] Department of Veterinary Pathology and Microbiology, Faculty of Veterinary Medicine, Universiti Putra Malaysia, Serdang, Selangor, Malaysia
[4] Department of Veterinary Clinical Studies, Faculty of Veterinary Medicine, Universiti Putra Malaysia, Serdang, Selangor, Malaysia

## ABSTRACT

*Salmonella enterica* subsp. *enterica* serovar Stanley (*S.* Stanley) is a pathogen that contaminates food, and is related to *Salmonella* outbreaks in a variety of hosts such as humans and farm animals through products like dairy items and vegetables. Despite the fact that several vaccines of *Salmonella* strains had been constructed, none of them were developed according to serovar Stanley up to this day. This study presents results of genome sequencing and analysis on our *S.* Stanley UPM 517 strain taken from fecal swabs of 21-day-old healthy commercial chickens in Perak, Malaysia and used *Salmonella enterica* subsp. *enterica* serovar Typhimurium LT2 (*S.* Typhimurium LT2) as a reference to be compared with. First, sequencing and assembling of the *Salmonella* Stanley UPM 517 genome into a contiguous form were done. The work was then continued with scaffolding and gap filling. Annotation and alignment of the draft genome was performed with *S.* Typhimurium LT2. The other elements of virulence estimated in this study included *Salmonella* pathogenicity islands, resistance genes, prophages, virulence factors, plasmid regions, restriction-modification sites and the CRISPR-Cas system. The *S.* Stanley UPM 517 draft genome had a length of 4,736,817 bp with 4,730 coding sequence and 58 RNAs. It was discovered via genomic analysis on this strain that there were antimicrobial resistance properties toward a wide variety of antibiotics. Tcf and ste, the two fimbrial virulence clusters related with human and broiler intestinal colonizations which were not found in *S.* Typhimurium LT2, were atypically discovered in the *S.* Stanley UPM 517 genome. These clusters are involved in the intestinal colonization of human and broilers, respectively. There were seven *Salmonella* pathogenicity islands (SPIs) within the draft genome, which contained the virulence factors associated with *Salmonella* infection (except SPI-14). Five intact prophage regions, mostly comprising of the protein encoding Gifsy-1, Fels-1,

Corresponding author
Nurulfiza Mat Isa,
nurulfiza@upm.edu.my

RE-2010 and SEN34 prophages, were also encoded in the draft genome. Also identified were Type I–III restriction-modification sites and the CRISPR-Cas system of the Type I–E subtype. As this strain exhibited resistance toward numerous antibiotics, we distinguished several genes that had the potential for removal in the construction of a possible vaccine candidate to restrain and lessen the pervasiveness of salmonellosis and to function as an alternative to antibiotics.

# INTRODUCTION

According to *Crump et al. (2015)* and *Williamson et al. (2017)*, *Salmonella enterica* serovar Stanley (*S*. Stanley) is a zoonotic non-typhoidal pathogenic microorganism responsible for food contamination. Characterized as O:4 (B) serogroup with 1,4,[5],12,[27]:d:1,2 antigenic formula (*Grimont & Weill, 2007*), it is able to start salmonellosis outbreaks in humans by contaminating food-producing animals as well as eggs, dairy products, fruits and vegetables (*Dahshan et al., 2010*; *Eng et al., 2015*).

A few instances of salmonellosis outbreaks caused by *S*. Stanley had been reported before. The incidents were in Denmark and France in 2008. The Statens Serum Institute, Denmark, had found that *S*. Stanley was positioned the fifth most regular serovar which caused Salmonellosis in 3,657 cases. Meanwhile, the Institut Pasteur, France reported *S*. Stanley as being the 36th serotype that was related with human salmonellosis out of 10,378 serotyped *Salmonella*. During the period of 2002–2007, Thailand recorded 11% cases of salmonellosis in humans due to this serovar (*Hendriksen et al., 2009*, *2012*). In 2013 and 2014, the Center for Disease Control and Prevention (CDC) revealed cases of *Salmonella* infection due to *S*. Stanley involving 14 individuals from California, Nevada and Wyoming in the USA. Other than that, more recently similar cases were also reported in King County, Washington state, USA, infecting nine people in 2017 (*King County, 2017*).

No death had been reported so far, nevertheless, the implications of this outbreak included significant clinical symptoms and signs, for example, diarrhoea, abdominal cramps, vomiting, dehydration, nausea, muscle and joint pain as well as asthenia (*Pastore et al., 2008*). Hence, salmonellosis is still regarded as being of public health concern worldwide since it can affect the economies of a number of industrialized and developing nations and efforts are being made to put a stop to such outbreaks (*Hardy, 2004*; *Eng et al., 2015*).

Numerous measures have been put forward in the effort to lessen the spread of salmonellosis, such as by constructing antimicrobial agents. A number of studies which were done to know how susceptible *S*. Stanley was to antimicrobials revealed that isolates from humans, were highly resistant to ampicillin, chloramphenicol, streptomycin, sulfamethoxazole, trimethoprim and tetracycline but not so much to cephalosporins,

gentamicin, ciprofloxacin and nalidixic acid with different degrees of resistance (*Bagger-Skjøt et al., 2007*; *Huang et al., 2007*). Apart from that, *Dahshan et al.'s (2010)* study described that *S.* Stanley in bovine species appeared to be resistant to multi-drugs including ampicillin, chloramphenicol, streptomycin, sulfamethoxazole, oxytetracycline, trimethoprim and kanamycin. The existence of multi-drug resistance is of significant concern to global public health authorities due to the fact that it can possibly lessen the potency of the antibiotics. Thus, more appropriate intervention measures needed to be applied in order to efficiently decrease the spread of *S.* Stanley. One of the measures includes constructing vaccines that can fight against antimicrobial resistance while lowering the pathogenic and virulence features of *S.* Stanley (*Clift & Salisbury, 2017*). Several *Salmonella* vaccines for poultry have been made which come in different forms such as spray, oral administration and so on (*Varmuzova et al., 2016*; *Methner et al., 2004*). But until today, not one of the vaccines developed used serovar Stanley as a base. Each of the serovars possesses a distinct virulence determinant that is able to trigger the immune response.

A review was done by *Blakeway et al. (2017)* on a number of virulence determinants of a pathogenic bacterium in the human nasopharynx, *Moraxella catarrhalis*. It is crucial to study factors that determine the virulence of *Moraxella catarrhalis* in order to construct a vaccine to control the pervasiveness of the diseases caused by this pathogenic bacterium including otitis media in children as well as chronic obstructive pulmonary disease in adults. In addition, a sequencing of *Leptospira interrogans* serovar Copenhageni strain Fiocruz L1–130 was conducted by *Gamberini et al. (2005)* to determine vaccine candidates of vaccine that might be effective against the cause of human leptospirosis: *Leptospira*. *Wren (2000)* explained that conducting sequencing on the entire genome of pathogenic microorganisms is a way to develop weakened vaccines. In order to construct weakened vaccines that are economical, this task can be accomplished through the finding and removal of virulence genes.

Thus, in order to better understand the genomic characteristics of *S.* Stanley's virulence and pathogenicity, we performed sequencing as well as analysis on our *S.* Stanley UPM 517 draft genome. The goal was to determine the elements associated with the virulence of *S.* Stanley UPM 517 that might be deleted to develop a potential vaccine. The strain of *S.* Stanley UPM 517 from fecal swabs isolated in random from 21-day-old healthy commercial broiler chickens in Perak which were submitted to the Faculty of Veterinary Medicine, UPM for diagnosis, had been subjected to sequencing and annotation. The reference genome used for this study was *Salmonella enterica* subsp. *enterica* serovar Typhimurium str. LT2 (*S.* Typhimurium LT2), a significantly related and thoroughly studied complete genome. A number of analyses were performed on *S.* Stanley UPM 517, namely determination of resistance gene, pathogenic island, virulence factor, prophage, plasmid region, restriction-modification site and the CRISPR-Cas system. The results we obtained could help in explaining the pathogenic features of serovar Stanley which would allow potential vaccines to be developed to lessen the prevalence of *Salmonella* outbreaks as well as to act as a substitute to antibiotics.

## MATERIALS AND METHODS

### Genome sequencing and de novo assembly

A strain sample of *S.* Stanley UPM 517 from fecal swabs of 21-day-old healthy commercial broiler chickens in Perak, Malaysia was sent to the Faculty of Veterinary Medicine, UPM. The Illumina Hiseq Technology was used to sequence the genome and the CLC Genomics Workbench version 7.5.1 (CLC Bio, Qiagen, Valencia, CA) with $16.7\times$ of genome coverage was used for the assembly work.

BLASTn analysis was performed on the draft genome to get the reference genomes for scaffolding. *Salmonella enterica* subsp. *enterica* serovar Yovokome str. S-1850 (accession no.: CP019418.1), *Salmonella enterica* subsp. *enterica* serovar Typhimurium strain LT2 (accession no.: AE006468.2), strain CDC 2009K-2059 (accession no.: CP014983.1), strain USDA-ARS-USMARC-1896 (accession no.: CP014977.1) and strain USDA-ARS-USMARC-1898 (accession no.: CP014971.2) were chosen as their characteristics relate the most with the reference genomes (*E*-value = 0.0, identity and query cover = 99%, max score $\geq 3.837e^{+05}$ and total score $\geq 6.817e^{+05}$). MeDuSa (*Bosi et al., 2015*) and GapBlaster (*De Sá et al., 2016*) were, respectively, used for scaffolding and gap filling to further improve the *S.* Stanley UPM 517 contiguous sequences. The linker sequences (NNNNN) were added to connect the scaffolds and to construct a "pseudogenome."

### Gene prediction and functional annotation

The RAST server was used to annotate the *S.* Stanley UPM 517 draft genome (*Aziz et al., 2008*) in which it identified potential genes and regions of coding via Glimmer (*Kelley et al., 2012*). On the other hand, RAST used the tRNAscan-SE (*Lowe & Eddy, 1996*) to determine tRNA and the Niels Larsen's "search_for_rnas" to determine rRNA. The annotated draft genome of *S.* Stanley UPM 517 had been deposited in the European Nucleotide Archive with accession number GCA_900834415. Homology searching by using BLAST was performed on *Salmonella* Stanley UPM 517 to get a complete genome that was thoroughly studied and closely related to *S.* Stanley UPM 517. *Salmonella enterica* subsp. *enterica* serovar Typhimurium str. LT2 (accession no.: NC_003197.2) which had been isolated by *McClelland et al. (2001)* was chosen as a reference genome with 0.0 of *E*-value and 99% of identity. Analysis on the orthologs (COGs) cluster was done by using OrthoVenn (*Wang et al., 2015*) with default parameters (*E*-value: 1e-5 and inflation value: 1.5).

### Whole genome alignment

The *S.* Stanley UPM 517 that had been annotated was aligned with the complete genome of *S.* Typhimurium LT2 with default parameters of progressiveMauve. The average nucleotide identity (ANI) calculated by referring to MUMmer was done by using JSpeciesWS (*Richter et al., 2015*).

### Identification of resistance genes and *Salmonella* pathogenicity islands

Detection of acquired antimicrobial resistance genes was done using resistance gene identifier from the CARD database (*Jia et al., 2017*). Low levels of resemblance hits of genes were then removed. The identified resistance genes were validated via in vitro antimicrobial

susceptibility testing which was carried out for six antimicrobials, namely ampicilin, cefotaxime, nalidixic acid, polymyxin B, tetracycline and trimethoprim–sulphamethoxazole. SPIFinder, which can be obtained from the Center for Genomic Epidemiology website (http://www.genomicepidemiology.org/), was used to identify *Salmonella* pathogenicity islands (SPIs) with 95% and 60% of identity threshold and minimum length, respectively. Then, BLAST analysis was performed on the predicted SPIs against the virulence factor database (VFDB) (*Chen et al., 2005*) for identifying the existence of virulence factors in each SPI which included bacterial pathogens, toxins and virulence genes.

### Prediction of virulence factors, prophages and plasmid region

With *E*-value of 1.0e-10, the same database was used to predict the virulence factors of the entire draft genome. On the other hand, PHASTER (*Arndt et al., 2016*) was used to identify the sequence of prophages contained in the genomes. Regions that scored above 90 were known as intact prophage, regions that scored 70–90 as questionable prophage and regions scoring below 70 as incomplete prophage (*Arndt et al., 2016*). Next, VFDB was used to analyze the prophages that had been identified to find the presence of possible virulence genes within their regions. PlasmidFinder was then used to identify plasmid regions of the draft genome (*Carattoli et al., 2014*), and for optimized results, it was also performed on PATRIC BLAST (*Wattam et al., 2014*) against the plasmid contigs database.

### Determination of restriction-modification sites and CRISPR

Draft genome analysis was also carried out using a prediction tool constructed from the curated database REBASE (*Roberts et al., 2015*), Restriction-ModificationFinder (https://cge.cbs.dtu.dk/services/Restriction-ModificationFinder/) for identification of restriction-modification sites. Analysis on CRISPR regions was also performed using CRISPRCasFinder (*Couvin et al., 2018*). The lowest level of evidence, 1, was used for the CRISPR regions with one to three spacers while the levels of evidence 2–4 were subjected to the other regions which were considered as true CRISPR. A similar tool with "Subtyping" precision level was used to identify the associated proteins next to CRISPR (Cas proteins). The spacers between CRISPR arrays that had been determined were subsequently assigned to CRISPRTarget (*Biswas et al., 2013*) for the identification of potential protospacers against the Genbank-Phage and Refseq-Plasmid databases with cutoff score $\geq 20$.

## RESULTS

### General genome features

Table 1 shows the *S*. Stanley UPM 517 and *S*. Typhimurium LT2 genome features in brief. The sequence of *S*. Stanley UPM 517 genome draft had an estimated length of 4,736,817 bp with GC content of 52.2% and 4,730 coding sequences (CDSs). As is shown in Fig. 1, *S*. Stanley UPM 517 was detected to contain 54 tRNAs and four rRNAs which included two 5S rRNAs, one 16S rRNA and one 23S rRNA.

**Table 1 General genome characteristics of *S. Stanley* UPM 517 and *S. Typhimurium* LT2 genome.**

| Features | *S. Stanley* UPM 517 | *S. Typhimurium* LT2 |
|---|---|---|
| Status | Draft | Complete |
| Genome size (bp) | 4,736,817 | 4,857,432 |
| GC content (%) | 52.2 | 53.0 |
| Contigs | 94 | – |
| Scaffolds | 9 | – |
| Total no. of CDS | 4,730 | 4,489 |
| Total no. of tRNA | 54 | 85 |
| Total no. of rRNA | 4 | 22 |
| References | This work | *McClelland et al. (2001)* |

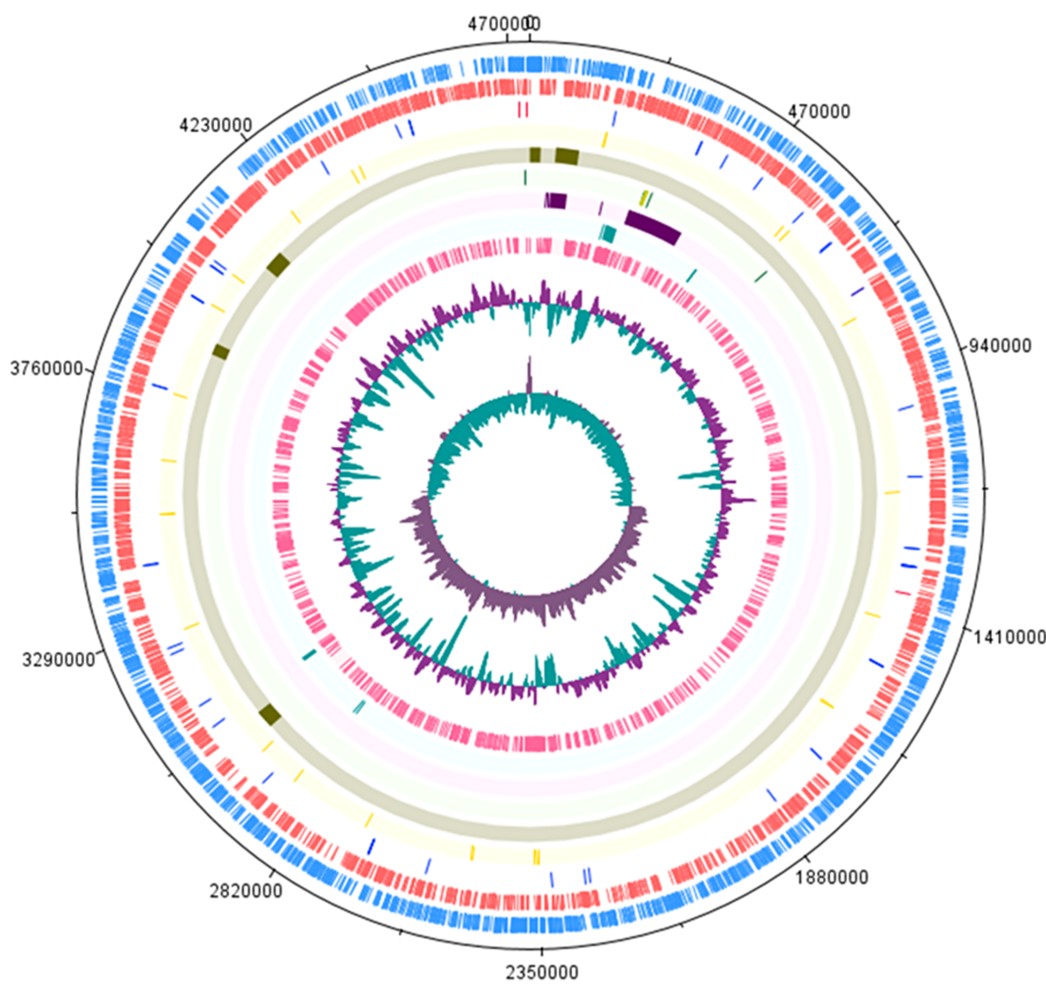

**Figure 1 Circular representation of *S. Stanley* UPM 517 draft genome.** Tracks from the outermost are as follows: Forward CDS, reverse CDS, RNA (blue represents tRNA and red represents rRNA), SPIs, virulence factors, CRISPR-Cas system (green indicates CRISPR region and yellow indicates cas genes), plasmid regions, antimicrobial genes and prophage regions. The two inner tracks are G+C content and GC bias (innermost track). The circular map was generated by using DNAPlotter (*Carver et al., 2009*).

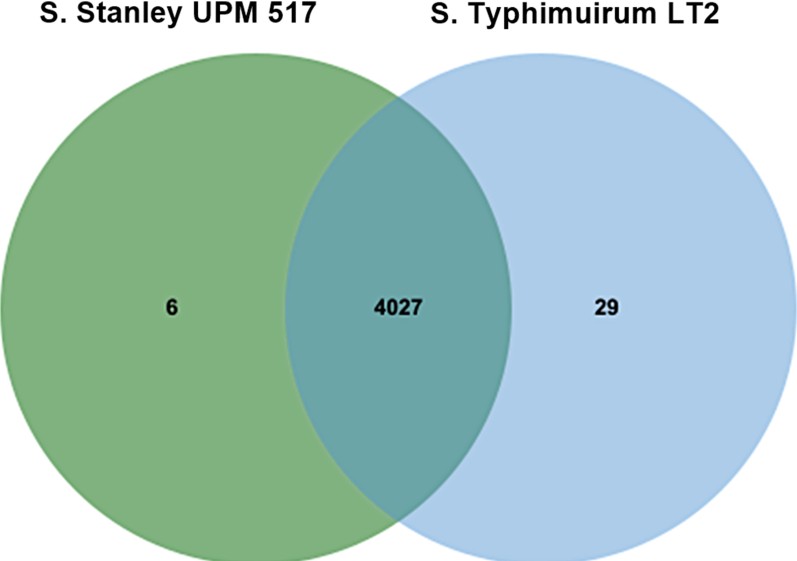

**S. Stanley UPM 517**   **S. Typhimuirum LT2**

**Figure 2 Unique and shared COGs between S. Stanley UPM 517 and S. Typhimurium LT2.** Both strains shared higher number of COGs, showing that they carry similar functional capabilities.

## Cluster of orthologs groups

It was found that 4,027 out of the 4,033 COGs contained in *Salmonella* Stanley UPM 517 were also present in *S*. Typhimurium LT2 (Fig. 2). Six atypical orthologous clusters contained in the *Salmonella* Stanley UPM 517 genome consisted of 12 protein encoding carnitine transporter 7, uncharacterized mitochondrial protein ymf21 and hypothetical proteins.

## Whole genome alignment

Using Mauve's multiple genome alignments, it was found that the regions of *S*. Stanley UPM 517 and of *S*. Typhimurium LT2 were highly conserved (Fig. 3), but two locally collinear blocks of *S*. Stanley UPM 517 were reversed. However, *S*. Stanley UPM 517 and *S*. Typhimurium LT2 exhibited the highest identity (98.97%) as shown by the ANI calculation prediction based on MUMmer by JSpeciesWS.

## Antimicrobial resistance genes

It was predicted that *S*. Stanley UPM 517 consisted of 27 genes coded to be resistant to a number of antibiotics such as fluoroquinolone, penicillin, macrolide, fosfomycin, peptide, aminocoumarin, aminoglycoside, monobactam, carbapenem, rifamycin, triclosan, glycylcycline, tetracycline, cephalosporin, phenicol, cephamycin, penems, carbapenem rhodamine, benzalkonium chloride, acycline, streptogramin and nitroimidazole (Table S1), among which, 16 were identified as having multi-drug resistance abilities. The major mechanism of the resistance against antibiotics is the efflux pump. It was found that *S*. Stanley UPM 517 was resistant against each of the six antibiotics (Table 2) based on the test of antimicrobial resistance via in vitro validation.

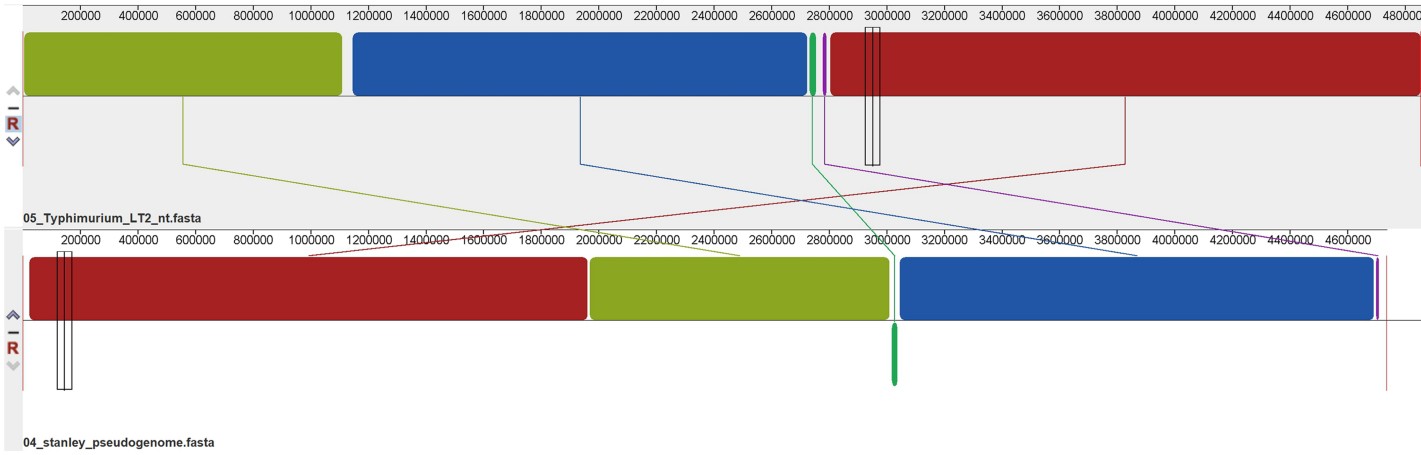

**Figure 3** **Whole genome alignment between *S.* Stanley UPM 517 and *S.* Typhimurium LT2 by progressive Mauve.** The lines connecting between blocks indicate the blocks were conserved between both *S.* Stanley UPM 517 and *S.* Typhimurium LT2 genome. A slightly sequence rearrangement and inversed block was also presented. The inversed block indicates reverse complement orientation of *S.* Stanley UPM 517 corresponds to *S.* Typhimurium LT2 genome.

**Table 2 Antimicrobial susceptibility testing of *S.* Stanley UPM 517.**

| Antibiotic | Zone diameter breakpoint (mm) | | | Results (mm) |
|---|---|---|---|---|
| | S≥ | I | R≤ | |
| Ampicilin (amp) | 15 | – | 14 | 10.0 |
| Cefotaxime (ctx) | 30 | 24–29 | 23 | 14.0 |
| Nalidixic acid (na) | 18 | – | 17 | 11.0 |
| Polymyxin B (pb) | 15 | 15 | 14 | 7.5 |
| Tetracycline (te) | 24 | – | 23 | 9.0 |
| Trimethoprim–sulphamethoxazole (sxt) | 17 | 14–16 | 13 | 13.5 |

**Note:**
The diameter of inhibition zones were measured and were categorized according to the following categories: S, susceptible; I, intermediate; R, resistant. *Salmonella* Stanley UPM 517 was resistant toward all the tested antibiotics.

## Virulence factors and *Salmonella* pathogenicity islands

A minimum of 994 (20.96%) out of the 4,730 CDS annotated in *S.* Stanley UPM 517 were homologous to the virulence factors outlined in VFDB (Table S2). It was discovered that *S.* Stanley UPM 517 contained two fimbrae clusters, namely tcf that carried tcfABCD genes and ste that carried steABCDEF genes. However, these were not found in *S.* Typhimurium LT2. In addition, it was revealed that the *S.* Stanley UPM 517 genome contained seven *Salmonella* pathogenicity islands, namely SPI-1, SPI-2, SPI-5, SPI-6, SPI-13, SPI-14 as well as centisome 63 pathogenicity island (C63PI). A total of 35 virulence factors encoding the required proteins for invasion of cells and secretion were carried in *S.* Stanley UPM 517's SPI-1 (Fig. 4A). A manual check on the VFDB results revealed two virulence factors clusters encoding for SPI-2 and SPI-6 (Figs. 4B and 4C). The SPI-5 of the same genome was also connected with six virulence factors that were involved in translocation, namely pipB, ipgE, sopB/sigD, abpB, phoR and mprA proteins (Fig. 4D). Meanwhile, iron transporter proteins (sitABCD) that encode for C63PI of *S.* Stanley UPM

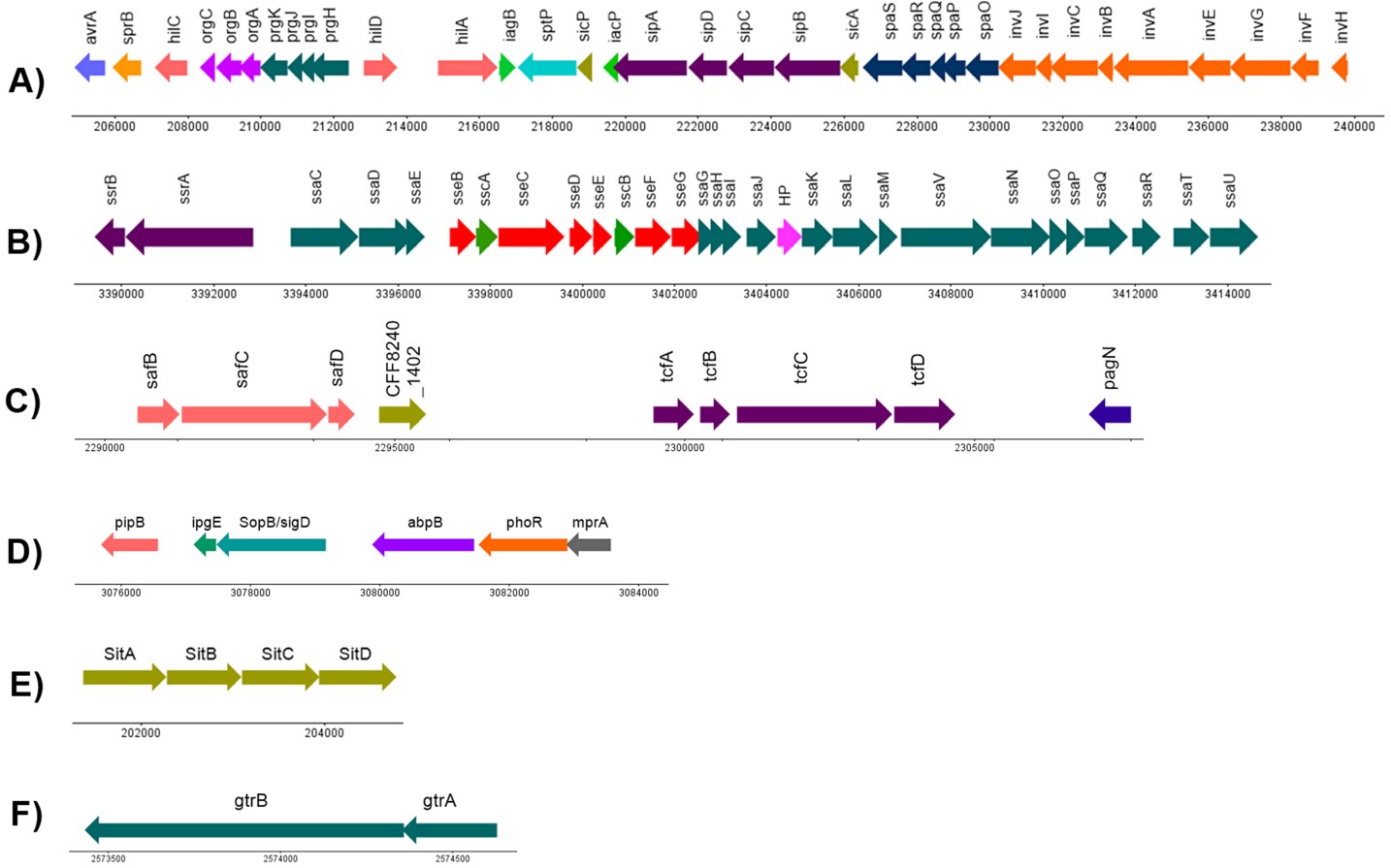

**Figure 4 Genetic organization of virulence factors in (A) SPI-1, (B) SPI-2, (C) SPI-6, (D) SPI-5, (E) C63PI and (F) SPI-13.** HP indicates hypothetical protein. The linear map was generated by using DNAPlotter (*Carver et al., 2009*).

517 genome (Fig. 4E), and gtrA and gtrB genes were discovered in SPI-13 (Fig. 4F). However, SPI-14 did not carry any of the virulence factors.

## Prophage regions

Table S3 shows the identified prophages found within the DNA sequences, flanking between the attL and the attR attachment sites. Five regions of intact prophages were contained in *Salmonella* Stanley UPM 517 with a range of 24.0–50.1 Kb and 46.80–51.64% of sizes and GC content, respectively (Fig. 1). The *S.* Stanley UPM 517 and *S.* Typhimurium LT2 genomes both had Gifsy-1 and Fels-1 conserved. Although not present in *S.* Typhimurium LT2, a great amount of protein-encoding SEN34 and RE-2010 prophages were, respectively, present in abundance in region 1 and region 2. Most of the ORFs in region 1 encoded for the hypothetical protein, but some had predicted functions including tail fiber protein, ClpA-like protein and portal protein. A number of phage-encoded virulence factors were contained in the prophage regions as well, such as GogB protein which was present in prophage region 3, FljA in prophage region 2 and EspO1-1 in prophage region 4.

**Table 3 CRISPR regions identified in *S.* Stanley UPM 517 genome and its properties.**

| CRISPR locus | Evidence level | DR length (bp) | No. of spacers | Conservation spacer | Start | End |
|---|---|---|---|---|---|---|
| CRISPR 1 | 4 | 29 | 9 | 0% | 273,426 | 274,003 |
| CRISPR 2 | 4 | 29 | 25 | 0% | 290,135 | 291,688 |
| CRISPR 3 | 1 | 26 | 1 | 100% | 614,013 | 614,115 |
| CRISPR 4 | 1 | 25 | 1 | 100% | 4,725,915 | 4,725,991 |

**Note:**

CRISPR locus with evidence level 1–2 are considered as invalid candidates while locus with evidence level 3–4 are assigned as highly predicted candidates. Both CRISPR 1 and CRISPR 2 contained 29 bp of direct repeats (DR). Cas cassette was found between CRISPR 1 and CRISPR 2 locus, locating from 274,100 to 282,550 bp of the whole genome.

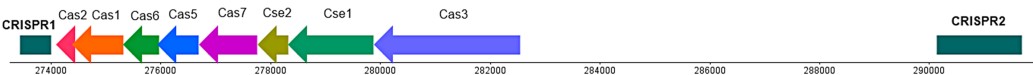

**Figure 5 The distribution of CRISPR-Cas system in *S.* Stanley UPM 517 genome.** The genetic organization of CRISPR and its associated protein represents Type I–E subsytem.

## Plasmid region

There was no result obtained from identifying plasmid via PlasmidFinder, so an alternative approach as explained by *Edwards & Holt (2013)* known as the BLAST algorithm was carried out in this analysis. The results of PATRIC BLAST hits showed that five plasmid regions displayed high resemblance to *S.* Stanley UPM 517 regions, namely plasmid:2 and plasmid:4 of *S.* Seftenberg, pKp5-1 of *Klebsiella pneumoniae* KP5-1, pESI of *Salmonella* Infantis 119944 and p34978 of *Enterobacter cloacae*, each of which, respectively, showed 97%, 98%, 95%, 91% and 93% of identity resemblance.

## Restriction-modification and CRISPR-Cas system

A total of three R-M systems (RMSs) were found in *S.* Stanley UPM 517 comprising of Type I, Type II and Type III. Cleaving was expected at the recognition motif GACNNNNNNNRTCC (the nucleotides underlined represented m6 sites) for Type I methyltransferase, M.Sen3124II; Type II methyltransferase, M.SenAboDcm and M.Sen1175III, recognized CCWGG (the nucleotides underlined represented m5 sites) and ATGCAT (the nucleotides underlined represented m6 sites) motifs, respectively. Meanwhile, Type III restriction enzymes, M.Sen8391II as well as SenAZII, respectively recognized at CAGAG (the nucleotides underlined represented m6 site) and CAGAG motifs. However, the cleavage site of SenAZII was unknown.

Table 3 shows the results which summarized the CRISPR region predictions by CRISPRCasFinder. Based on the results, it was discovered that *S.* Stanley UPM 517 contained two CRISPRs at evidence of level 4 and also two at the lowest level of evidence. CRISPR 1 and CRISPR 2 displayed direct lengths of 29 bp, each with, respectively, 9 and 25 unique spacers, while the CRISPR 1 and CRISPR 2 loci were both found next to CRISPR-associated helicase, Cas2, Cas1, Cas6, Cas5, Cas7, Cse2, Cse1, Cas3 (Fig. 5). Meanwhile, the other CRISPR loci which only possessed one spacer each and Cas protein

**Table 4 Identified protospacers within CRISPR arrays of *S.* Stanley UPM 517 with their scores.**

| CRISPR | Spacer | Prophage/plasmid | Accession number | Score |
|--------|--------|------------------|------------------|-------|
| CRISPR 1 | Spacer 2 | Plasmid pSF83666a of *Sinorhizobium fredii* strain CCBAU 83666 | NZ_CP023072.1 | 20 |
| | Spacer 4 | Plasmid pOXA10_020023 of *Escherichia coli* strain SCEC020023 | NZ_CP025944.2 | 32 |
| | | Plasmid pCFR-9161 of *Citrobacter freundii* complex sp. CFNIH3 | NZ_CP026237.1 | 32 |
| | | Plasmid unitig_2 of *Escherichia coli* strain AR_0114 | NZ_CP021734.1 | 32 |
| | | Plasmid unamed2 of *Pantoea ananatis* strain 97-1 | NZ_CP020945.1 | 20 |
| | | *Gordonia* phage Eyre | KX557277 | 20 |
| | | Plasmid pDSJ10 of *Pantoea stewartii* subsp. *stewartii* DC283 | NZ_CP017591.1 | 20 |
| | Spacer 8 | Plasmid byi_1p of *Burkholderia sp*. YI23 | NC_016626.1 | 20 |
| | | Plasmid p1 of *Burkholderia sp*. RPE67 | NZ_AP014579.1 | 20 |
| | | Plasmid pkk1 of *Burkholderia sp*. KK1 | NZ_CP016002.1 | 20 |
| | | Plasmid AbAZ39_p1 of *Azospirillum brasilense* strain Az39 | NZ_CP007794.1 | 20 |
| CRISPR 2 | Spacer 3 | Plasmid unnamed1of *Azospirillum sp*. M2T2B2 | NZ_CP029830.1 | 20 |
| | Spacer 4 | *Cronobacter* phage ESSI-2 | HQ110083 | 20 |
| | Spacer 8 | Plasmid:2 of *Salmonella enterica* subsp. *enterica* serovar Seftenberg NCTC10384 | NZ_LN868944.1 | 32 |
| | Spacer 9 | *Erwinia* phage Pavtok | MH426726 | 20 |
| | | *Erwinia* phage PEp14 | JN585957 | 24 |
| | | *Pseudomonas* phage phiAH14a | KU708004 | 24 |
| | Spacer 13 | Plasmid pLS1-1 of *Pseudonocardia sp*. HH130629-09 | NZ_CP011869.1 | 20 |
| | Spacer 18 | *Thiobacimonas* phage vB_ThpS-P1 | KT381864 | 20 |
| | | Bacteriophage P27 | AJ298298 | 20 |
| | Spacer 21 | Plasmid unnamed of *Pseudomonas monteilii* strain FDAARGOS_171 | NZ_CP014062.1 | 20 |
| | Spacer 23 | Plasmid punamed2 of *Salmonella enterica* subsp. *enterica* serovar Quebec str. S-1267 | NZ_CP022019.1 | 20 |
| CRISPR 3 | Spacer 1 | Plasmid: 2 of *Salmonella enterica* subsp. *enterica* serovar Senftenberg NCTC10384 | NZ_LN868944.1 | 45 |
| | | Plasmid punamed2 of *Salmonella enterica* subsp. *enterica* serovar Quebec str. S-1267 | NZ_CP022019.1 | 43 |

could not be found nearby these potential CRISPRs. A total of 12 out of 36 spacers were recognized as protospacers (Table 4), eight and three of which were identified as homologs to plasmid sequences and phage sequences, respectively. In addition, one spacer (Spacer 4) in CRISPR 1 was identified, harboring within both plasmid and phage.

## DISCUSSION

### Genome features

In an attempt to study bacterial factors, the *S.* Stanley UPM 517's genome sequence was obtained from fecal swabs of 21-day-old healthy commercial broiler chickens in Perak, Malaysia. Table 2 shows in general, the genome characteristics of both *S.* Stanley UPM 517 and of *S.* Typhimurium LT2. It was discovered that the size of the *Salmonella* Stanley UPM 517 genome was rather small, possibly resulting from incomplete genome sequencing. In addition, there was a lower GC content level in *S.* Stanley UPM 517 than in *S.* Typhimurium LT2 by a small difference of 0.8%, which indicated the homogeneity relationship of these strains. Apart from that, there was a substantial amount of intersecting orthologous clusters (4,040 clusters) in *S.* Stanley UPM 517 with

*S*. Typhimurium LT2 which showed that these two strains might have the same functional capabilities.

## Whole genome alignment

Alignment of the genome was also carried out on each strain (Fig. 3). The two genomes appeared to have shared the majority of blocks which indicated a substantial amount of genetic information conserved between the strains. The predicted value of ANI between *S*. Stanley UPM 517 and other strains showed the same gene-content contained in the strains and thus they were highly related to one another despite coming from different serovars.

## Plasmid regions

Draft genome sequences usually contain plasmid sequences which act as mediators responsible for sending physiological traits to the bacterial host and causing bacterial evolution and adaptation to the environment (*Shintani, Sanchez & Kimbara, 2015*). They also usually contain genes for antimicrobial resistance and virulence in *Salmonella* resulting in pathogenic features (*García et al., 2014*; *Carattoli et al., 2014*; *Silva, Puente & Calva, 2017*). Plasmid:2 and plasmid:4 of *S*. Senftenberg NCTC10384 strain greatly resembled the *S*. Stanley UPM 517's regions. Based on these results, a conjugative transfer of plasmid:2 and plasmid:4 might have taken place between *S*. Stanley UPM 517 and *S*. Seftenberg strain NCTC10384 since these two had a similar environmental niche.

## Antimicrobial resistance genes

A number of previous researches on multi-drug resistant *S*. Stanley had discovered that *S*. Stanley was resistant to some antibiotics such as ampicillin, chloramphenicol, streptomycin, sulfamethoxazole, trimethoprim, tetracycline, cephalosporins, gentamicin, ciprofloxacin, nalidixic acid, oxytetracycline and kanamycin to different extents (*Bagger-Skjøt et al., 2007*; *Huang et al., 2007*; *Dahshan et al., 2010*). The prediction analysis that we conducted on resistance genes showed that the *S*. Stanley resistance phenotype against antibiotics had come into existence, thus widening the range of antibiotics that this strain is resistant to.

## Virulence factors and SPIs

Since the *S*. Stanley UPM 517 is resistant to a wider range of antibiotics, a practical way to lessen the pervasiveness of *Salmonella* outbreaks is required, for example, vaccination. According to *Desin, Köster & Potter (2013)* and *Revolledo & Ferreira (2012)*, a great number of *Salmonella* vaccines had been developed by producing mutations or removing virulence determinants. An attenuated CVL30 derivative had been constructed by *Cooper et al. (1994)* using mutated aroA gene as a reference which would interfere with the key protein production in the aromatic biosynthetic pathway. Likewise, a *S*. Gallinarum oral vaccine had been developed by *Penha Filho et al. (2010)* by removing cobS and cbiA genes that disrupted cobalamin synthesis. After being tested, this vaccine was found to be effective toward *S*. Gallinarum and *S*. Enteritidis serovars. Also developed was a vaccine of live attenuated *S*. Typhimurium which had its cya and crp genes deleted. It was

discovered that the vaccine managed to significantly reduce the challenge strain level in the ceca and could not be traced in the chicken spleen but was somehow ineffective toward *S.* Enteritidis (*Hassan & Curtiss, 1994*). Thus, examining the characteristics of *S.* Stanley UPM 517's virulence is important to obtain a vaccine that is potent against a vast array of *Salmonella*.

The key roles of virulence factors include bacterial adherence, invasion, replication and survival within the host cells (*Ribet & Cossart, 2015*). SPIs are the particular regions in which several virulence factors are contained (*Kaur & Jain, 2012*). SPIs are usually found close to tRNA with a distinct GC content. The SPIs in *Salmonella* probably gained its virulence factors via horizontal gene transfer (HGT), a characteristic that permits *Salmonella* to manifest its virulence purpose and which helps to infect the host cell with salmonellosis (*Hensel, 2004*). The prediction analysis on SPIs showed that SPI-1, SPI-2, SPI-5, SPI-6, SPI-13, SPI-14 and C63PI were associated within the draft genome of *S.* Stanley UPM 517 which accomplished the task that we wanted done.

The virulence factor structures in SPI-1 for both *S.* Stanley UPM 517 and *S.* Typhimurium LT2 were conserved (Fig. 4A). It is known that SPI-1 in *Salmonella enterica* encodes for invasion, secretion system and translocation of proteins from the extracellular of *Salmonella* into the host cells (*Gerlach & Hensel, 2007*). In the *S.* Stanley UPM 517 draft genome, SPI-1 was found to contain a number of effector proteins such as hilA, sipA, sipB, sptP, invA and invG. *Bohez et al. (2007)* discovered that a mutation of the SPI-1's major transcriptional regulator, hilA, which was responsible for the regulation of these effector proteins, could be a potential vaccine candidate for broilers against salmonellosis. They reported that invasion was successfully reduced by inoculating the non-invasive hilA mutant to the 1-day-old chickens. This inhibited colonization of the challenged strain in the intestines after 24 h and this continued for 9 days upon oral administration. Another study conducted by *Matulova et al. (2012)* reported the removal or deletion of the entire SPI-1 (SPI-1 mutant) and administered as an oral vaccine to 1-day-old chickens were able to block the colonization of the wild type *S.* Enteritidis, hence, protecting the poultry from the challenged strain. *Dieye et al. (2009)* obtained a similar finding in their study where 1-day-old chickens which had been infected with *S.* Typhimurium strain of SPI-1 mutant (the whole SPI-1 removed) successfully reduced the colonization of bacteria in the chicken's cecum and spleen.

An effector protein, sipA, as discovered by *Galkin et al. (2002)*, *Higashide et al. (2002)*, *Zhou, Mooseker & Galán (1999a)* and *(1999b)*, was bound to the actin to modulate the host-cell small GTPases and to regulate the signal transduction pathways. This functional manner subsequently activated F-actin polymerization at the site of *Salmonella* adhesion and membrane ruffles formation, thus facilitating the efficiency of *Salmonella* uptake. Other effector proteins, invA and invG, were involved in the invasion of *S. typhimurium* to the epithelial cells as reported by *Galan, Ginocchio & Costeas (1992)* and *Kaniga, Bossio & Galán (1994)*. *Kubori & Galán (2003)* and *McGhie et al. (2009)* reported that an effector protein sptP could survive longer in the host cell after the invasion and it was responsible for terminating membrane ruffling and repaired the invaded-induced damage by reversing the inflammatory mechanisms to ensure *Salmonella* intracellular survival in

the invaded cell. Another study conducted by *Chaudhuri et al. (2013)* demonstrated that the sptP mutant in *S.* Typhimurium ST4/74 could reduce the colonization on chicks, pigs and cattle's intestine. Meanwhile, *Hernandez et al. (2003)* suggested an effector protein sipB played an important role in disrupting mitochondria and caused macrophage apoptosis, thereby leading to cell death. Thus, all the above-mentioned effector proteins identified in *S.* Stanley UPM517 could be considered for removal to enable it to be a potential vaccine candidate.

Both *S.* Stanley UPM 517 and *S.* Typhimurium LT2's virulence factor distributions and organizations were conserved in SPI-1. However, three virulence factors encoding for SPI-2, namely ssaB (spiC), sseA and ssaS were not present in *S.* Stanley UPM 517. Studies performed by *Ruiz-Albert et al. (2003)* and *Coombes et al. (2003)* found that effector proteins such as sseB and sseD could not be translocated in the mutant sseA strains. They hypothesized that sseA might act as a chaperone required for the translocation of the SPI-2-encoded effector proteins to the host cells. Similarly, *Freeman et al. (2002)* discovered that ssaB (spiC) was needed to translocate the SPI-2-encoded effector proteins such as sseB and sseC. However, in the *S.* Stanley UPM 517 genome, translocon proteins (sseB, sseC and sseD) were still present in the absence of sseA and ssaB. This might indicate that the translocations of sseD and sseC of *S.* Stanley UPM 517 to the host cell were not dependent on sseA and ssaB, respectively, but might be dependent on sseB translocon. This finding was supported by a study performed by *Chakravortty et al. (2005)* who found that sseC and sseD proteins were dependent on sseB translocon as both sseC and sseD were not secreted by the sseB mutant in *S.* Typhimurium. Further studies must be conducted to validate the functionality of SPI-2 of *S.* Stanley UPM 517 and its proteins in depth. Another study focusing on intracellular survival capacity of *S.* Stanley UPM 517 in macrophages may also need to be performed to investigate whether the absence of ssaB, sseA and ssaS in SPI-2 would significantly affect the survival of *S.* Stanley in the host cells. Alternatively, ssaV in SPI-2 might also be considered for deletion before being used as a potential vaccine candidate as previously *Figueira et al. (2013)* showed that the mutation of this gene attenuated *S.* Typhimurium.

The sopB/sigD effector protein of SPI-5 also promoted the Salmonella invasion and membrane ruffling of the host cell in a phosphoinositide phosphatase manner and it was co-regulated together with the SPI-1 effector proteins and secreted by T3SS of SPI-1 (*Knodler et al., 2002*). On the other hand, the pipB of SPI-5 was released and translocated to the *Salmonella*-induced filaments and *Salmonella*-containing vacuole upon infection, through T3SS of SPI-2 (*Knodler et al., 2002*). It was also described in a previous research that pipB of *S.* Typhimurium played a role in chicken enteropathogenesis (*Morgan et al., 2004*). Interestingly, the virulence factors that were earlier not related to any SPIs, namely mprA and abpB, were discovered in the SPI-5 of *S.* Stanley UPM 517. Besides, two virulence factors, mprA and abpB, which were previously not associated with any SPIs, were predicted to be found in SPI-5 of *S.* Stanley UPM 517. Based on the results listed from VFDB, these two virulence factors formerly originated from *Mycobacterium vanbaalenii* and *Streptococcus gordonii*, respectively, suggesting that these virulence factors might have undergone HGT events which allowed the transfer of mprA and abpB to

*Salmonella* from other genera. According to *He & Zahrt (2005)*, mprA participated in the regulatory system to promote and maintain the growth of *Mycobacterium tuberculosis* at the persistent stage of infection within the host. Meanwhile, the abpB protein which encoded for amylase binding protein, might play a role in *Salmonella* Typhimurium LT2 infection and colonization by either degrading gylcan or by interacting with other effector proteins to establish biofilm formation (*Arabyan, Huang & Weimer, 2017*; *Chaudhuri et al., 2008*).

The interesting finding was that *S.* Stanley UPM 517 appeared to have two uncommon fimbrial gene clusters, tcf and ste, which explained its distinctive virulence characteristics and therefore made it easier to understand the host-interaction of the *S.* Stanley UPM 517 genome. The collection of these fimbrial clusters might also give bits of knowledge on the construction of a vaccine candidate as they were exposed to the immune system. In *S.* Enteritidis, the ste cluster comprised of six fimbrial genes and played a role as a major cluster associated with chicken intestine colonization (*Clayton et al., 2008*). Initially, the tcf cluster was only discovered in typhoidal *Salmonella*, *S.* Typhi and *S.* Paratyphi A, and in a few non-typhoidal serovars like Choleraesuis, Heidelberg, Schwarzengrund, Virchow, Typhisuis, Muenchen and Montevideo (*Townsend et al., 2001*; *Bronowski & Winstanley, 2009*). It had previously been described as contributing to human host colonization and more recent studies had been carried out in an effort to figure out features of its functional roles (*Leclerc et al., 2016*). Comprising of four virulence genes which encoded for chaperone (tcfA), major subunit (tcfB), usher (tcfC) and adhesin (tcfD), it was inclusively discovered in SPI-6 alongside the saf cluster and pagN. A study by *Pezoa et al. (2013)* discovered that the SPI-6 mutation had reduced the gut and internal organ colonizations in chicks. Meanwhile, it was found in another study by *Carnell et al. (2007)* that the reduction of gastrointestinal colonization in pigs was caused by the mutation of the saf cluster. Two regions in the *S.* Stanley UPM 517 draft genome were found to be homologous to the SPI-13 and SPI-14 of *S.* Gallinarum. This result demonstrated that SPI-13 and SPI-14 were possibly being transferred from a phage or plasmid through HGT. Two genes, gtrA and gtrB, were discovered in the SPI-13 of *S.* Stanley UPM 517 (Fig. 4). These genes encoded for bactoprenol-linked glucose translocase and bactoprenol glucose transferase, respectively, in the modification of O-antigen structure (*Davies et al., 2013*). Nonetheless, none of ORFs were found in SPI-14 and thus its function was not thoroughly characterized.

Several SPIs were found missing in the *S.* Stanley UPM 517 draft genome, namely SPI-3, SPI-4, SPI-16 and SPI-9, but the important virulence factors that encoded them were detected such as SPI-3-encoded (misL), SPI-4-encoded (siiE), SPI-16-encoded (safBCD) and SPI-9-encoded (rpoS) which facilitated the adhesion of *Salmonella* to the host. CS54-encoded virulence factors such as shdA, ratB and sinH were found near to one another in the *S.* Stanley UPM 517 draft genome. Besides that, the fimbrial biosynthesis-encoded operons (csg, fim, lpf, bcf, saf, stb, stc, std, sth and sti) were also presented. Both CS54-encoded and fimbrial biosynthesis-encoded operons are responsible for colinization of mice intestine and facilitating the bacterial intestinal persistence in the mice, respectively (*Weening et al., 2005*).

## Prophage regions

The viruses that infect bacteria are called prophages or bacteriophages (*Sharma et al., 2017*). They are functionally associated with the bacterial lifestyle, fitness, virulence, evolution and pathogenicity (*Fortier & Sekulovic, 2013*; *Fu et al., 2017*). Gifsy-1 is a lambda-related prophage found in the *S.* Stanley UPM 517 draft genome which is predicted to play a role in the *S.* Stanley UPM 517 strain virulence since it was found to be harboring gogB, a Type III secretion virulence factor. However, GipA, an extensive phage encoded *S.* Typhimurium's virulence factor was absent in the Gifsy-1 of *S.* Stanley UPM 517. This is presumably a disadvantage to serovar Stanley since it is required for the survival of Peyer's patches (*Stanley, Ellermeier & Slauch, 2000*). However, the reported salmonellosis cases caused by serovar Stanley were still continuously occurring in several countries, which downgraded the significance of GipA for virulence in the hosts.

Fels-1 combined with Gifsy-1 prophage would create an improved bacterial virulence which when triggered might transfer their genetic materials to the host to get a better chance at survival. Such factors being distributed in *Salmonella* might assist in the construction of new pathogens (*Garcia-Russell, Elrod & Dominguez, 2009*). The *Salmonella* Stanley UPM 517 genome was also found to contain RE-2010 prophage which had been shown by a previous study to possess high nucleotide identity with the Fels-2 prophage (*Colavecchio et al., 2017*). It was also believed to function similarly as Fels-2 and was combined with Gifsy-1 and Fels-1 prophages in *S.* Stanley UPM 517. In addition, it was also found that a number of genes encoding SEN34 phage were contained in *S.* Stanley UPM 517. Having close connection with *Siphoviridae* phages, SEN34 phage is an unfamiliar temperate phage belonging to the *Myoviridae* family in terms of morphology (*Mikalová et al., 2017*). However, its significance in the *Salmonella* host is still unclear.

## Restriction-modification and CRISPR-Cas systems

The important role of the RMS in bacteria is to fight invading DNA by providing immune defence mechanisms via methylation activity performed on its own DNA (*Vasu & Nagaraja, 2013*). There were four types of R-M, namely Types I, II, III and IV (*Roer et al., 2016*), all of which were present in the *S.* Stanley UPM 517 genome except for Type IV. The CRISPR-Cas system plays a significant role in the bacterial adaptive immune system (*Barrangou & Marraffini, 2014*; *Burmistrz & Pyrć, 2015*; *Rath et al., 2015*) and its combination with the RMS improves the bacteriophage's resistance level (*Dupuis et al., 2013*). It was found that the *S.* Stanley UPM 517 strain contained four CRISPR loci of 36 spacers each, 12 of which were protospacers identified in plasmids and phages (Table 4). These findings revealed that *S.* Stanley UPM 517 had the ability to generate immunity against several phages and plasmids. In addition, each of the 36 spacers was atypically identified in the *S.* Stanley UPM 517 genome which indicated rapid transformation of CRISPR 1 and CRISPR 2. In this study, an evaluation was conducted to identify the existence of Cas genes which are important for CRISPR activity as they encode for the crucial proteins of the immune system (*Rath et al., 2015*). They are usually found close to the CRISPR sites and work alongside crRNA by

integrating the spacers into the CRISPR locus to improve bacterial immunity (*Rath et al., 2015*; *Hille & Charpentier, 2016*). The CRISPR-Cas system's structure and distribution from *S.* Stanley UPM 517 belonged to Type I–E subtype (*Makarova et al., 2011*). The structure of the CRISPR-Cas system was also present in the *S.* Typhimurium LT2 genome (*Fabre et al., 2012*). Perhaps the introduction of LacI to *S.* Stanley UPM 517 could be used as a vaccine candidate since according to *Eswarappa et al. (2009)* and *Louwen et al. (2014)*, the introduction of LacI repressor could significantly induce the expression of the cas genes (cas1, cas2, cas3, cas5, cas6 and cas7) and subsequently reduce the virulence of *S. enterica* by disrupting the expression of the virulence genes in SPI-2.

## CONCLUSION

In conclusion, we have singled out several genes that we found to be suitable for removal in the development of potential vaccine candidates for controlling and reducing salmonella pervasiveness as well as to become a substitute to antibiotics. Removing these genes might help with the construction of a vaccine, such as a live attenuated one, to fight against *Salmonella*. In addition, based on the comparisons made with the reference genome, we also found that our strain had the potential to colonize and spread through broilers as well as humans as it possessed the tcf and ste clusters. Apart from that, we also discovered that despite not having the secretion apparatus in SPI-2, this strain managed to survive being in the host. There is a need for further studies to measure the capacity of the intracellular survival of this serovar in the macrophages. More research on comparative genomics involving a vast variety of serovars should also be conducted to further characterize the currently little known serovar Stanley.

## ACKNOWLEDGEMENTS

We would like to express our thanks to the Institute of Bioscience and Faculty of Biotechnology and Biomolecular Sciences for providing the computational facilities as well as to everyone who have extended their direct or indirect support toward this work. The authors would also like to thank Professor Soon Guan Tan, formerly Associate Editor of Elsevier Editorial System, Gene, for proof reading the manuscript.

### Funding
This work was supported by the FRGS (No. FRGS/1/2017/STG05/UPM/02/18), Putra Grant IPS (No. 9622500) and IBS HICoE (No. 6369101) from the Ministry of Higher Education, Government of Malaysia. The funders had no role in study design, data collection and analysis, decision to publish, or preparation of the manuscript.

### Grant Disclosures
The following grant information was disclosed by the authors:
FRGS: FRGS/1/2017/STG05/UPM/02/18.

Putra Grant IPS: 9622500.
IBS HICoE: 6369101.
Ministry of Higher Education, Government of Malaysia.

## Competing Interests

The authors declare that they have no competing interests.

## Author Contributions

- Khalidah Syahirah Ashari conceived and designed the experiments, performed the experiments, analyzed the data, contributed reagents/materials/analysis tools, prepared figures and/or tables, authored or reviewed drafts of the paper.
- Najwa Syahirah Roslan conceived and designed the experiments, performed the experiments, analyzed the data, contributed reagents/materials/analysis tools.
- Abdul Rahman Omar contributed reagents/materials/analysis tools, authored or reviewed drafts of the paper, approved the final draft.
- Mohd Hair Bejo contributed reagents/materials/analysis tools, authored or reviewed drafts of the paper, approved the final draft.
- Aini Ideris contributed reagents/materials/analysis tools, approved the final draft.
- Nurulfiza Mat Isa conceived and designed the experiments, performed the experiments, contributed reagents/materials/analysis tools, authored or reviewed drafts of the paper, approved the final draft.

## DNA Deposition

The following information was supplied regarding the deposition of DNA sequences:

The sequence data of *Salmonella enterica* subsp. *enterica* serovar Stanley UPM 517 are available at the European Nucleotide Archive under accession number GCA_900834415.

## Data Availability

The raw scaffold sequences and embl flat file are available in the Supplementary Files.

## Supplemental Information

Supplemental information for this article can be found online at http://dx.doi.org/10.7717/peerj.6948#supplemental-information.

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
