# Peer review of "Genome sequencing and analysis of Salmonella enterica subsp. enterica serovar Stanley UPM 517: Insights on its virulence-associated elements and their potentials as vaccine candidates"

_PeerJ, doi:10.7717/peerj.6948_

## Round 0.1 · original submission · Major Revisions

Dear Dr. Ashari and colleagues:

Thanks for submitting your manuscript to PeerJ. I have now received three independent reviews of your work, and as you will see, the reviewers raised some concerns about the research. Nonetheless, these is enough optimism for me to encourage you to revise your work and resubmit. Importantly, please ensure that an English expert thoroughly evaluates your revision. Also, the quality of the scientific writing is low, so please comply with the suggestions raised by the reviewers to improve the the writing and presentation style.

Regarding scientific content, in your revision please describe in detail the major differences between this serovar and closely related serovars. The absence of the SPI 2 secretions system (and anticipated effects of a lack of secreted effectors) should be discussed in considerable detail. Per reviewer 3, please downplay all usage of “virulence factors” as you have not characterized virulence in this report. I think the title should also be modified as suggested by reviewers 1 and 3.

Accordingly, I am recommending that you revise your manuscript, taking into account all of the issues raised by the reviewers. I look forward to seeing your revision, and thanks again for submitting your work to PeerJ.

Good luck with your revision,

-joe

·

Basic reporting

1. The language needs some polishing e.g.
a. ‘Two fimbrial virulence clusters, tcf and ste, which were absent in S. Typhimurium LT2, were uniquely distributed in S. Stanley UPM 517 genome.’ – what does uniquely distributed mean? Uniquely found?
b. ‘has shown resistant towards many antibiotics’ – you mean ‘shown resistance’
c. ‘There is no death reported yet, however, this outbreak results in serious clinical signs and symptoms, such as diarrhea, abdominal cramps, vomiting, dehydration, nausea, muscle and joint pain and also asthenia (Pastore et al., 2008). Severe gastrointestinal colonization may occur, leading to severe systemic infection and can cause death (Guiney, 2005)’ – so, no known deaths but death could occur? You should remove the ‘Severe gastro…’ sentence.
d. ‘Our aim is to characterize the virulence associated-elements of S. Stanley UPM 517 draft genome to become a potential vaccine candidate’ – you mean ‘to identify a potential vaccine candidate’

2. It is not clear from your language what work was done here and was done previously
a. ‘These clusters were associated with human and broilers intestinal colonization respectively.’ – where are these data presented? Or is this from previous work?

3. The raw sequence data is not available. NQWD00000000.1 links to the assembled (unannotated) contigs. You should upload your raw sequence data (European Nucleotide Archive is easier than NCBI) and you definitely have to upload your annotated genome, as this is the whole subject of the paper.

4. In the discussion, please give some examples of genomics being used for the rational design of vaccine candidates?

5. ‘highly homologous’ you cannot say highly homologous, something either is homologous or it is not. Like being unique. Replace with just homologous, or ‘highly similar’.

6. In the discussion, it is sometimes difficult to determine what you have done and what is from the literature.
a. ‘These effector proteins were regulated by a major transcriptional regulator of SPI-1, hilA.’ You don’t show any data concerning this, so you should say ‘The effector proteins are regulated’ and then give the reference.
b. Another example, you need to provide a reference or make it clear that this info is from the literature - ‘SopB/sigD effector protein of SPI-5 also promoted the Salmonella invasion and membrane ruffling of the host cell in phosphoinositide phosphatase manner.’

Experimental design

1. How was this isolate selected for sequencing? Random? Because of the large number of antibiotic resistance genes?

Validity of the findings

1. Would be good to have some phenotypic antimicrobial resistnace testing to help validate the in silico findings.
2. Why do you have more coading sequences in UPM than in LT2? Despite the fact that your assembly is > 100 kbp shorter. Please address in the text.
3. Genome alignment – do you think it is valid to compare your genome versus LT2 when you have re-arranged your draft genome vs LT2? If you want to talk about genome re-arrangements, you have to show me the contigs with the re-arrangement in. When you map reads back to that contig, do the reads support the re-arrangement, or is it possibly a mis-assembly?
4. ‘Of these resistance genes, 16 genes encode for multi-drug resistant genes’ what is a multi-drug resistant gene? A gene which gives resistance to multiple drugs?
5. L203 ‘Two unique clusters of fimbrae’, unique compared with what?
6. ‘five functional prophage regions’ how do you know that they were functional? You haven’t provided your annotated genomes so I can’t judge for myself whether the pro-phage were intact or not.
7. You need to provide more details of the potential plasmids you identified? How much of the reference plasmids were present in your Stanley assembly?
8. ‘These results indicated that S. Stanley UPM 517 was able to provide immunity against a number of foreign nucleic acids such as viruses and plasmids’ – this review says that there is no convincing evidence of recent spacer acquisition in Salmonella, did you find something different? https://www.ncbi.nlm.nih.gov/pubmed/25479838

Additional comments

Ashari et al have carried out a detailed examination of the genome of a Salmonella Stanley isolate from broilers. The work appears largely technically sound, and is very thorough. The authors do a good job of discussing the genes which could be knocked out in potential vaccine candidates. However, I have numerous issues with the manuscript, detailed in the provided sections. The most pressing issue is that the annotated genome has not been made publicly available.

Some of the language in the manuscript is potentially misleading and this needs a lot of attention. One example is the title:

'In silico identification of potential vaccine candidate of Salmonella... '

You have not identified a potential vaccine candidate, this appears to be a normal, virulent Salmonella Stanley. You have identified genes which should be considered for removal from a potential vaccine candidate. The title therefore requires modification.

Reviewer 2 ·

Basic reporting

See comments in "General comments for the author" section

Experimental design

See comments in "General comments for the author" section

Validity of the findings

See comments in "General comments for the author" section

Additional comments

In this study the uthors report the draft genome sequence of a broiler-isoleted strain of Salmonella Stanley. From this data, the authors propose some potential vaccine candidate to control the pathogen, which frequently causes outbreaks to humans.

First, the quality of writing is not suitable for publication unless extensively edited, making the paper very hard to read. I strongly suggest the authors to use a language editing service for their manuscript.

Several researchers have tried to develop vaccines through the use of attenuated strains. Given that the authors of this study ultimately discuss a similar approach, it is not clear why it was necessary to obtain a complete genomic sequence for this: it would have been sufficient to obtain mutants in the virulence factors. In other words, it is worth asking what is the novelty that can be deduced from the data about the biology of this serovar. The authors should discuss in more depth what is the real contribution that, eventually, makes this serovar different from the others.

Probably the most interesting finding of the study is the lack of the secretory apparatus encoded in the SPI 2. A question that arises from this is how the isolate could survive in the animal without the protection provided by the effectors secreted by this system of serción. A simple experiment to perform that could give information about it, would be to measure the intracellular survival capacity of the bacteria in macrophages.

I am sure the authors put serious efforts in this study. However, I feel that the manuscript deserves an equal attention.

Reviewer 3 ·

Basic reporting

In the manuscript "In silico identification of potential vaccine candidate of Salmonella enterica subsp. enterica serovar Stanley UPM 517 draft genome sequence and its virulence-associated elements" the authors have sequenced the genome of a strain isolated from a broiler chicken and have performed computational analysis in order to identify potential new virulence factors, genomic and pathogenicity islands, antimicrobial resistance genes and CRISPR/Cas systems in this bacteria.

The manuscript is well-written, scientifically sound and suitable for publication in this journal. It also adds interesting data to the Salmonella community. I have only minor comments before the manuscript is suitable for publication in PeerJ.

Experimental design

no comment

Validity of the findings

My only comments and suggestions are the following:

1) The current title "In silico identification of potential vaccine candidate of Salmonella enterica subsp. enterica serovar Stanley UPM 517 draft genome sequence and its virulence-associated elements" overstates the main findings of the manuscript and my suggestion is to modify it to avoid confusion. The current work corresponds more to the genome sequencing and analysis study of this bacterial strain rather than a systematic study to identify vaccine candidates in silico. A more suitable title would be: ¨Genome sequencing and analysis of Salmonella Stanley UPM 517a: insights in its virulence-associated elements and potential vaccine candidates”.

2) The same applies to the main conclusion (Line 45-446) “our study revealed that the virulence-associated elements identified in S. Stanley could potentially be used and modified as candidates for vaccination to control and reduce the prevalence of salmonellosis as well as an alternative to antibiotics”. I think the language can be softened highlighting the main aspects of the paper without overstating and overreaching its main findings.

---

## Round 0.2 · Minor Revisions

Dear Dr. Ashari and colleagues:

Thanks for re-submitting your manuscript to PeerJ. I was able to obtain two reviews from the three original reviewers of your work, and as you will see, one reviewer still raised some concerns about the research. The other is still not satisfied with the language edits, though the English does appear much improved in the revision. Please continue to work on the grammar. Nonetheless, there is enough optimism for me to encourage you to revise your work and resubmit again. Importantly, please ensure that the reviewer’s concerns about the scientific writing are addressed, especially regarding the literature cited. Also, please be sure to specifically define the bacteria to which you refer to in the appropriate sections.

Accordingly, I am recommending that you revise your manuscript, taking into account all of the issues raised by the reviewer. I look forward to seeing your revision, which I am sure will be ready for publication once these concerns are met. Thanks again for submitting your work to PeerJ.

Good luck with your revision,

-joe

Reviewer 2 ·

Basic reporting

The writing is still deficitary.

Experimental design

OK

Validity of the findings

See the general comments

Additional comments

This is a substantially modified version of the manuscript, containing a big amount of changes concerning the scientific writing quality as well as in diverse aspects of the results description and discussion. Respect to writing, it is certain that quality was improved, but it is still rough, so in occasions the messages can be misunderstood or misinterpreted. Below, you can find some concerns from the discussion section, some of which eventually could be due to that, concerns that should be addressed.

A first general concern is about the descriptions of the evidences from literature. Frequently the description does not include the specie or serovar from which the evidence was generated, leading to the reader to think that the evidence could be reffered to the Stanley serovar. The authors should be more rigorous about that. For instance, see lines 416-418.

A great amount of writing concerns were detected. Here you have two of them.
Line 437-439. “The other putative SPI which is associated with intestinal colonization of the host is CS54-encoded virulence factors (shdA, ratB and sinH) were discovered to be close together in S. Stanley UPM 517” Please revise the redaction
Line 439-441 “Also presented was fimbrial biosynthesis-encoded operons (csg, fim, lpf, bcf, saf, stb, stc, std, sth and sti) and was said to contribute to the persistence of bacterial intestinal in the host (Weening et al., 2005)”. Please revise the redaction

Major concerns
lines 291-292. "This is possibly because there were more intergenic regions presented in S. Typhimurium LT2". Do the authors want to say a higher number of intergenic regions? If it is so, this statement would not necessarily explain why UPM 517 has a higher number of coding sequences than LT2 strain, in spite of the genome of UPM 517 has a slightly lower size than the LT2 one: this explanation does not exclude the possibility that the higher number of intergenic regions, the higher number of genes

353-355. "It is noted that SPI-1 in Salmonella is responsible for encoding of invasion, secretion system and translocation of proteins to host cells from Salmonella's extracellular".

356-359. "SPI-1's major transcriptional regulator, hilA, is responsible for the regulation of these effector proteins (Bohez et al., 2007) and was initially found to be a potential vaccine for broiler as discovered by Bohez et al. (2007) that SPI-1 was able to protect broilers against salmonellosis" So, the authors state that HilA could be used as a vaccine against S. Stanley. Considering the below, I think that they wanted to say taht a hilA mutant could be a candidate for a vaccine.

line 362-365." According to another study by Matulova et al. (2012), the entire SPI-1 was removed and via oral administration, it was used as a vaccine to 1 day-old chickens. This deletion has revealed that the wild-type Salmonella was able to block colonization, hence protected the poultry from the challenged strain". WT bolcks colonization or the colonization of wt was blocked?

line 365-368. "Dieye et al. (2009) obtained similar finding in their study where 1 day-old chickens had been infected with the strain of SPI-1 mutant (the whole SPI-1 removed) Dieye et al. (2009) obtained similar finding in their study where 1 day-old chickens had been infected with the strain of SPI-1 mutant (the whole SPI-1 removed) with a resistant towards several antibiotics such as nalidixic acid, chloramphenicol and kanamycin, and successfully reduced the colonization of bacteria in the chicken's cecum and spleen. and successfully reduced the colonization of bacteria in the chicken's cecum and spleen. " Why the authors included the resistant profile in this statement?... it is confusing and unnecessary for stating the point.

370-373. "The binding of an effector protein, sipA, to the actin in modulating the small GTPases of the host cell and regulating the signal transduction pathways subsequently resulted in polymerization of F actin being activated at the bacterial adhesion sites as well as formation of membrane ruffles, hence improving the Salmonella uptake effectiveness". To my undestanding this statement is not referred to serovar Stanley, as no experiments to study this system were conducted in this work. So, the authors should clearly indicate which bacteria they refer and some reference(s) should be included.

line 382-392. This is a controversial paragraph. First, some references should be included to support that "It was shown in previous research that the effector proteins of the mutant ssaB and sseA strains could not be translocated". Second, which is the "protector" activity of SseA. Third, the authors stated that "Nonetheless, even without sseA and ssaB, the translocon proteins like sseB and sseD remained available, indicating that sseA has no influence over the translocation of the S. Stanley UPM 517's effector proteins to the host cell." What are the evidences that support the statement... this is highly speculative. Finally, the authors propose removal ssaV gene as a vaccine candidate, but as previously they stated, in the absence of ssaB and sseA effector proteins cannot be secreted, so it is probable that secretory system encoded in SPI2 was not functional. As I stated in the previous report, a question that arises from this is how Stanley could survive in the animal without the protection provided by the effectors secreted by this system. Has this serovar a functional TTSS-2?. This should be discussed.

line 394-396 "The effector protein of SPI-5, SopB/sigD, also aids in the invasion of Salmonella and host cell's membrane ruffling in phosphoinositide phosphatase way and co-regulates alongside the effector proteins of SPI-1 and secreted by T3SS of SPI-1." This is a very confusing redaction.

Line 400-401. “In addition, the virulence factors that were earlier not related to any SPIs, namely mprA and abpB, were expected to be discovered in SPI-5 of S. Stanley UPM 517”. Why the authors expect this possibility?

Lines 428-432. “ORFs in S. Stanley UPM 517 draft genome was found to be homologous to the SPI-13 and SPI-14 in S. Gallinarum. It was showed from the result that both SPI-13 and SPI-14 might have been transferred from a plasmid or phage via HGT. Nonetheless, the functions of SPI-13 and SPI-14 in Salmonella pathogenicity had yet to be thoroughly characterized”. What would be the orfs present in these islands?. It is confusing, because the paragraph is referred to fimbrial genes cluster and as shown in figure 4 this islands, al least the SPI-13, do not included them.

Lines 451-453. “Nonetheless, the virulence role of GipA in hosts may be arguable as a number of countries were reported to still having cases of salmonellosis continuously.” This statement seems to be out of context. The authors should be clearly indicate the serorovarto which they refer to.

Line 467-485. How these genetic elements could contribute to serovar Stanley virulence and the eventual vaccine development?

Reviewer 3 ·

Basic reporting

The use of english language has improved from the original version of the manuscript but there are still many parts throughout the manuscript that need to be revised. The authors claim that an editing service was used but that is not reflected on the manuscript.

Experimental design

My previous concerns were taken into account. I have no further comments.

Validity of the findings

My previous concerns were taken into account. I have no further comments.

---

## Round 0.3 · accepted · Accept

Dear Dr. Ashari and colleagues:

Thanks for once again revising your manuscript based on the minor concerns raised by the reviewer. I now believe that your manuscript is suitable for publication. Congratulations! I look forward to seeing this work in print, and I anticipate it being an important resource for researchers studying Salmonella pathogenesis. Thanks again for choosing PeerJ to publish such important work.

Best,

-joe